# Long-Range Language Modeling with Selective Cache

**Xinting Huang**
University of Copenhagen
wbn969@alumni.ku.dk

**Nora Hollenstein**
University of Copenhagen
nora.hollenstein@hum.ku.dk

## Abstract

The computational cost of transformer-based language models grows quadratically with the sequence length. In this paper, we introduce the selective cache, which stores the selected key-value pairs from the previous context. By selecting important key-value pairs the model makes better use of the cache so that in limited cache size, a longer context history can be stored. We design three kinds of selection methods. The first is based on human language processing. The key-value pairs are selected if they correspond to tokens that are fixated longer, as recorded in eye-tracking-while-reading experiments. We also incorporate the cognitively-inspired selection process into the language model as a trainable process, resulting in two additional methods with improved performance. The selection task is converted into a pruning task so they can be trained with differentiable masks. We demonstrate that the proposed selective cache improves the language modeling performance across different datasets.[1] With the same number of stored key-value pairs (cache size), our selective cache outperforms XL cache (Dai et al., 2019) and compressive cache (Rae et al., 2019) by considerable margins.

## 1 Introduction

Transformers (Vaswani et al., 2017) have been used as the backbone architecture of various models and achieve state-of-the-art results in a wide range of tasks (Devlin et al., 2019; Dosovitskiy et al., 2020; OpenAI, 2023; Kirillov et al., 2023). Compared to other architectures such as Recurrent Neural Networks, its attention mechanism directly connects long-distance elements in the input sequence. Thus, it better captures long-range dependencies. However, it also constitutes the bottleneck in Transformers. The time and space complexity of Transformer attention is $O(n^2)$. Hence, the computational cost becomes prohibitively expensive or even makes the task infeasible when processing long sequences. Extensive research is performed to address this problem, introducing a number of *X-former* models (Wang et al., 2020; Kitaev et al., 2020; Choromanski et al., 2022).

While many proposed models successfully mitigate the problem to some extent, there are still open limitations such as the storage cost of preserving the whole sequence in memory and the inability to be directly inserted into existing pretrained models. In this paper, we propose a novel cache that stores past key-value pairs, namely the *selective cache*, for auto-regressive language modeling (LM). This method aims to make better use of the cache introduced in Transformer-XL (Dai et al., 2019) by selecting more informative key-value pairs. Different from prior works such as (Rae et al., 2019), it distills the previous context by extracting or selecting, instead of averaging. In other words, some key-value pairs are kept in the cache as it is, while others are simply discarded. It is utilized as easily as the cache in Transformer-XL, that is, the stored keys and values are simply prepended to the current keys and values.

We explore three selection methods for the cache: (1) using eye fixation duration as importance scores, (2) using neural networks which learn to select, and (3) replace. The first one uses eye-movement data. We simply consider the total reading time of each word. Long-fixated words are important for language comprehension (Rayner and Duffy, 1986) and are usually difficult to infer from the context (Rayner et al., 2011). Furthermore, fixation duration correlates with the gradient-based saliency in Transformer language models (Hollenstein and Beinborn, 2021). Thus, selecting key-value pairs associated with long-fixated tokens is a cognitively plausible modeling decision. The second method is to train a selector neural network. By learn-

---

[1] Our code is available at https://github.com/huangxt39/SelectiveCacheForLM

ing to *prune* key-value pairs, the selector learns to *select*. It is based on a neural network pruning method (Louizos et al., 2017), which uses differentiable stochastic masks and $L_0$ penalty. The selector learns to mask out unimportant key-value pairs so it can be used to select them. Finally, the third method uses the same technique but applies the mask and its opposite mask simultaneously, in order to train a replacer network that directly learns the trade-off. The motivation is to simulate the actual situation where adding new items to the cache results in discarding old ones. In addition, we introduce a novel method for training the selector and replacer network. By sampling a previous snippet of context to train the network and stopping gradient during cache updating, we avoid backpropagation through time (BPTT), while the network can still learn to model long-term dependency.

Experimental results show that using these methods improves the model's performance, especially the latter two. They bring about considerable performance gain against XL cache and compressive cache (we use these two terms to refer to the methods used in Dai et al., 2019 and Rae et al., 2019, respectively) with the same total cache size.

Moreover, thanks to the enhanced interpretability of selective cache, we can directly investigate which tokens are stored. We find that the trained selector and replacer tend to keep named entities and rare tokens, which is a common tendency found in many prior works (Sun et al., 2021; Wu et al., 2022; Hutchins et al., 2022). We also show that the selective cache can preserve information from a very distant context, or even from an infinite distance, depending on the characteristics of the processed document.

## 2 Related Work

Transformers have difficulty processing long sequences. There are a lot of works trying to address this limitation. Linformer (Wang et al., 2020) and Performer (Choromanski et al., 2022) introduce new self-attention mechanisms, which can approximate the original full attention with linear space and time complexity. Many works utilize sparsity to achieve efficient long-range attention, including strided attention (Child et al., 2019; Beltagy et al., 2020), global attention by a few tokens (Beltagy et al., 2020; Zaheer et al., 2020; Ravula et al., 2020) and random attention to a limited number of tokens (Zaheer et al., 2020). Moreover, suitable

sparsity patterns can be learned for each attention head (Sukhbaatar et al., 2019; Correia et al., 2019). Some other works (Kitaev et al., 2020; Roy et al., 2021) use clustering techniques to partition elements in the sequence and perform intra-cluster attention. Besides, hierarchical structure (Zhu and Soricut, 2021; Ren et al., 2021) is incorporated into the attention mechanism to reduce computational complexity. Long documents can also be split into segments and processed in a recurrent manner. Block-Recurrent Transformer (Hutchins et al., 2022) use a set of hidden representations to store past information and use cross-attention to interact with them, while Memory Recurrent Transformer (Bulatov et al., 2022) utilizes dedicated memory tokens to do so.

Transformer-XL (Dai et al., 2019) achieves this segment-level recurrence by the cache mechanism, which stores the past hidden states and uses them as an extended context when the model processes the next input sequence. Compressive Transformer (Rae et al., 2019) extends the Transformer-XL with a secondary cache which compresses the old hidden states. On the other hand, Memorizing Transformer (Wu et al., 2022) expands the cache to an enormous size, and uses an approximate k-nearest-neighbor (kNN) search to retrieve key-value pairs from the cache efficiently. ∞-former (Martins et al., 2022) utilizes a continuous-space attention mechanism and represents the input sequence as a continuous signal, so that the long-term memory can be represented with lower precision. Expire-Span (Sukhbaatar et al., 2021) computes a span for each hidden state that determines how long it should stay in memory.

In this paper, we make use of eye-tracking data, which has been incorporated into many NLP frameworks (Mathias et al., 2020). Eye-tracking data provides advantages to NLP models in terms of both their performance and interpretability. Research has demonstrated that incorporating eye-tracking features can enhance prediction accuracy in tasks like named entity recognition (Hollenstein and Zhang, 2019; Tokunaga et al., 2017), part-of-speech tagging (Barrett et al., 2018), sentiment analysis (Mishra et al., 2017) and general NLP benchmark tasks (Khurana et al., 2023). In the meantime, eye-tracking data is utilized to explore the correlation between human behavior and neural attention (Hahn and Keller, 2023; Sood et al., 2020; Brandl and Hollenstein, 2022).

We also use a pruning technique based on stochastic masks (Louizos et al., 2017), which is mainly used in neural network pruning (Louizos et al., 2017) and interpretation (Voita et al., 2019; De Cao et al., 2020). Different from these approaches, we aim to improve performance by using it to select important parts of previous context. Even though the technique is used for different purposes in different scenarios, it actually does the same thing, i.e., learns what is less important.

# 3 Model

In this paper, we use decoder-only transformers (Vaswani et al., 2017) to perform auto-regressive LM tasks. Long documents are split into segments of 512 tokens. The segments are not shuffled and fed into the model sequentially. In other words, the language model processes the document step by step, one segment at a time, as is done in (Dai et al., 2019; Wu et al., 2022).

At each step, Transformer-XL (Dai et al., 2019) caches and fixes (stops gradient) the hidden state sequence computed for the current segment. In the next step, it is reused to extend the context. Although in (Dai et al., 2019), the hidden states are cached, we follow the practice in (Wu et al., 2022) and save the key-value pairs into the cache for the purpose of efficiency. When doing attention, the keys and values are prepended to the current keys and values. When the XL cache size $C_{xl}$ is greater than the input segment length $C_{inp}$, the XL cache is a first-in-first-out (FIFO) queue.

When using cached representations, it is necessary to use relative position embeddings. We use T5 relative position embeddings (Raffel et al., 2020), which adds different biases to different relative offsets when doing attention.

When switching to a new document, the cache may contain some content from the old document, we apply document masks to solve this problem. Concretely, for the cached key-value pairs, we keep track of their document IDs. Each token can only attend to other tokens with the same document ID.

These are the common settings used for all our experiments involving the cache, including baselines and proposed models.

## 3.1 Selective Cache

We aim to make better use of the cache by selecting those key-value pairs which are more beneficial than others. The selective cache is a FIFO queue

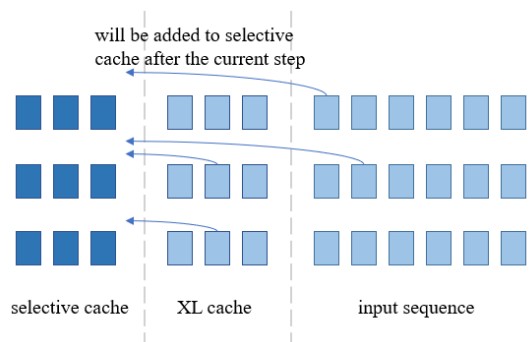

Figure 1: In the selective cache, we extend the attention layer with cached representations from previous context.

and is of a fixed size like XL cache. At each step, the key-value pairs that satisfy a certain criterion are selected and saved into the selective cache. In the meantime, the same number of old key-value pairs are discarded. Like XL cache, the selective cache is also non-differentiable. In this paper, all the models that use selective cache also use XL cache in the meantime. The selective cache serves as a secondary cache and selects tokens discarded by XL cache, like the compressive cache (Rae et al., 2019). So the model contains a detailed recent context, as well as a distilled context that covers a wider historical period. The cached keys and values from the selective cache are concatenated with those from XL cache and those from the current input. See illustration in Figure 1.

We also introduce a set of trainable bias parameters, the selective cache bias. They are used in the same manner as T5 position embeddings. Recall that T5 position embeddings are scalars added to the dot product of queries and keys. When the query attends to the keys in selective cache, the position embedding that corresponds to the maximum distance is used. Meanwhile, the selective cache bias (also a scalar) is added to the dot product as well. For each layer, there are $m$ bias parameters, where $m$ is the number of heads. So the same bias is used for all attention scores of one head.

## 3.2 Using Eye-fixation Duration as Criterion

A crucial component of the selective cache is the selection criterion. One plausible choice is human fixation data. While eye-tracking data contains various information about human eye movements during reading, we simply utilize the total reading time (TRT) of each word, which is the sum of all fixation duration on a single word. If an input token is associated with a TRT longer than a cer-

tain threshold, its corresponding key-value pairs are selected into the cache in all attention layers. The threshold is a hyperparameter that needs to be tuned. Because human reading is optimized for efficiency and accuracy, they only fixate words to the extent necessary for task success (Hahn and Keller, 2023). Long fixated words are usually important and are likely to contain new information which cannot be inferred from the context.

However, there is no available eye-tracking data for the LM datasets used in this paper. We choose to use a simple long short-term memory (LSTM) network to predict the fixation duration for the text in LM datasets. It is trained on the available eye-tracking corpora. After training it is able to predict the TRT within an acceptable error range (see Appendix B). The fixation prediction model only processes the text once at the preprocessing stage. The predicted fixation duration is used repeatedly for many epochs when training the language model.

### 3.3 Using Selector Network as Criterion

We introduce an automatic selector, which gives free rein to the model and let it decide which tokens should be cached. It is a small neural network integrated into the self-attention layer. The selection task is converted into a task similar to network pruning. The language model is provided with some previous key-value pairs when doing the LM task, the selector is encouraged to prune some of the key-value pairs. Its architecture is described in Appendix E.

**Binary Masks** For a sequence of past hidden states $[\vec{h}_1, \vec{h}_2, \cdots, \vec{h}_n]$ (the time index is omitted here for simplicity), the selector network takes the hidden states as input and outputs binary masks for each of them $[z_1, z_2, \cdots, z_n]$. The masks are used in self-attention as follows:

$$s_{ij} = \vec{q}_i \cdot \vec{k}_j + \ln z_j \tag{1}$$

where $s, \vec{q}, \vec{k}$ are the attention score before softmax, the query, and the key respectively. So when $z_j$ is zero, the corresponding key-value pair is masked out, when $z_j$ is one, the attention score is not affected. Note that here the position embeddings, document masks, and selective cache bias are all omitted for simplicity.

In the meantime, $L_0$ norm is applied to the pre-

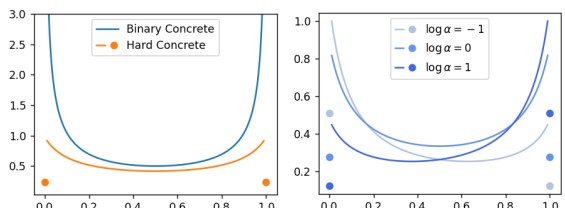

Figure 2: **Left:** The probability density function of Binary Concrete distribution and Hard Concrete distribution. In this example, $\log \alpha = 0, \beta = 0.5, \gamma = -0.1, \zeta = 1.1$, and $p(z = 0) = p(z = 1) = 0.23$. **Right:** Hard Concrete distribution with different $\log \alpha$ when $\beta = 0.4, \gamma = -0.1, \zeta = 1.1$.

dicted masks.

$$||z||_0 = \sum_{j=1}^{n} \mathbf{1}_{[\mathbb{R} \neq 0]}(z_j) \tag{2}$$

where $||z||_0$ denotes $L_0$ norm, $\mathbf{1}(\cdot)$ stands for indicator function. $L_0$ norm penalizes the number of non-zero values. It imposes a constant penalty everywhere except for $z_j = 0$. It encourages the selector to completely switch off some key-value pairs.

As we can see, $L_0$ norm is not a differentiable function and has zero derivative almost everywhere. Meanwhile, the outputs $z_j$ are not produced from a continuous function either.

**Stochastic Masks** The solution to these two problems is to use the stochastic masks drawn from some distributions controlled by the selector, and the $L_0$ norm becomes the *expected* number of non-zero masks. More specifically, we use Hard Concrete distribution (Louizos et al., 2017), which is a mixture of discrete and continuous distributions, ranging in the closed interval $[0, 1]$. It gives non-zero probability to the two endpoints $0, 1$, while between 0 and 1 the probability density function is continuous, as shown in Figure 2. The Hard Concrete distribution contains a few parameters, one of which is $\log \alpha$. It controls the probability mass skewing towards 0 or 1. See Appendix C for more information.

With the Hard Concrete distribution, the selector is trained as follows: On the one hand, the network outputs the parameter $\log \alpha_j$ for each input hidden state $\vec{h}_j$. Then one sample $z_j$ is drawn from each distribution and is used to mask attention scores as Equation 1. With the reparameterization trick (Kingma and Welling, 2013), the gradient can backpropagate through the sampling process

to the selector network. The training objective of language modeling would encourage the selector to output larger $\log \alpha$, so that extra information can be obtained to help predict the next token. On the other hand, $L_0$ norm is treated in expectation, which becomes a differentiable function of $\log \alpha$. It encourages the selector to output smaller $\log \alpha$.

With the influence of both aspects, important key-value pairs are likely to be kept. When the selector is used as the selection criterion, the $\log \alpha_j$ is compared with threshold 0. If $\log \alpha_j > 0$, $\vec{k}_j$ and $\vec{v}_j$ are put into the selective cache.

When using selector, the training loss becomes:

$$\mathcal{L} = \mathcal{L}_{LM} + \lambda \mathcal{L}_0 \qquad (3)$$

where $\mathcal{L}$ is the total loss, $\mathcal{L}_{LM}$ is the language modeling loss. $\mathcal{L}_0$ represents the $L_0$ norm. $\lambda$ is the coefficient. A key problem is to determine $\lambda$. We design an adaptive $\lambda$ as follows:

$$\lambda_t = r_t^2 \cdot \lambda' \quad r_t = r_{t-1}*0.9 + \frac{\#\text{selected}}{\#\text{total}}*0.1 \quad (4)$$

where $r_t$ is a running average of the selection ratio, $\lambda_t$ is the coefficient used at step $t$. $\lambda'$ is a hyperparameter. In this way, if the selection ratio grows too big, the "suppression force" also increases.

**Simultaneously Selecting and Training** The selective cache stores context of multiple previous steps, one way to train the selector is to backpropagate gradient through all relevant steps. We avoid this by separating the selecting and training process. We sample a random snippet of length $C_{snp} = 128$ from previous context to train the selector, i.e., perform the pruning task, while it also performs selection as a non-differentiable process. In other words, during training, some additional keys and values are prepended to the concatenation of selective cache, XL cache, and current keys and values. We control how distant the snippet is through a geometric distribution parameterized by a hyperparameter $l$. It should be determined based on how long the dependency we intend to capture. See more details in Appendix D.

### 3.4 Removal of Similar Instances

In preliminary experiments of training the selector, we find that some tokens are repeatedly selected. Because the selector network makes decisions on token-level, it does not have the overall information, e.g., which kind of information is already selected.

For this reason, it is possible that it selects very similar representations. To address this issue, we calculate the distance matrix of the selected values in each step. Values with a distance smaller than a certain threshold are considered in the same "cluster". Only one key-value pair in each "cluster" is finally selected. See more details in Appendix F. In this process, we only calculate the distance matrix once among the *selected* values. This process can be regarded as a secondary selection. In the following part of the paper, we refer to this technique as removal of similar instances (RSI).

### 3.5 Replacement-Based Selective Cache

When training the selector network, the $L_0$ penalty implies the capacity limit of the selective cache. Because the cache size is fixed, storing some new key-value pairs means discarding some old ones. Inspired by this process, we further propose a way to directly model this trade-off. Concretely, the key-value pairs are grouped two by two, a neural network compares the two key-value pairs and masks out either of them.

Suppose that the representations at position $j$ and $j'$ are compared. The approach calculates stochastic masks as follows.

$$\log \alpha_{jj'} = \mathrm{f}_{rpl}(\vec{h}_j, \Delta t_j) - \mathrm{f}_{rpl}(\vec{h}_{j'}, \Delta t_{j'}) \quad (5)$$
$$z_{jj'} \sim \mathrm{HardConcrete}(\log \alpha_{jj'}, \beta, \gamma, \zeta) \quad (6)$$

where $\mathrm{f}_{rpl}(\cdot)$ denotes the neural network trained (we refer to it as the replacer network, see its architecture in Appendix E), $\Delta t_j$ is the time interval between the current step and the previous step when processing position $j$. HardConcrete represents the Hard Concrete distribution, $\beta, \gamma, \zeta$ are constant parameters. Then the sampled random mask $z_{jj'}$ is used as follows.

$$s_{ij} = \vec{q}_i \cdot \vec{k}_j + \ln z_{jj'} \quad (7)$$
$$s_{ij'} = \vec{q}_i \cdot \vec{k}_{j'} + \ln(1 - z_{jj'}) \quad (8)$$

Therefore, ideally, only one of the two can be attended. Intuitively, if the key $\vec{k}_j$ and value $\vec{v}_j$ provide more important information for the current step, the replacer network should output a higher "score" $\mathrm{f}_{rpl}(\vec{h}_j, \Delta t_j)$ which results in high $\log \alpha_{jj'}$. The cost, on the other hand, is masking out $\vec{k}_{j'}$ and $\vec{v}_{j'}$.

However, this one-to-one relationship deviates from the actual situation. Adding one into the cache does not necessarily result in removing the other,

the removed key-value pair could be any one in the cache. Nevertheless, we still opt to use this relationship for its simplicity.

When training the replacer, there is no need for $L_0$ penalty. We again use a random snippet of previous context for training. A random set of key-value pairs in selective cache is chosen to do the one-to-one comparison with those in the snippet. The replacer network outputs masks, and each of the masks and its opposite mask are applied to one in the snippet and one in the selective cache respectively.

When updating, the replacement-based selective cache is not a first-in-first-out queue anymore. The input sequence is compared with the selective cache element-wise. The key-value pair with higher "score" $f_{rpl}(\vec{h}_j, \Delta t_j)$ is retained. Importantly, when switching to a new document, the key-value pair from the new document is always preserved no matter what the replacer outputs.

## 4 Experiments

We evaluate the models described above on three datasets using two different model sizes. The experiments are all language modeling and the performance is evaluated by perplexity per token. Besides, we also do various qualitative and quantitative analyses on the models. From these analyses, we provide strong evidence of the effectiveness of the proposed models.

### 4.1 Data

We introduce the language modeling datasets used in the experiments. The eye-tracking corpora are described in Appendix A

**PG-19** PG-19 language modeling benchmark is a LM dataset introduced by Rae et al. (2019). It includes a collection of English books published before 1919 from the Project Gutenberg books library. Each document in the PG-19 is a full-length book. However, due to limited computational resources, we only use 1% of the training set, which is 286 books, around 18M tokens. The documents are randomly selected and then fixed. So the same set of documents is used for all models. On the other hand, we use full validation and test set, which contain 3M and 7M tokens respectively. Even though only 1% is used, the training data still has a reasonable size and is enough for fine-tuning.

**WikiText-2** The WikiText language modeling dataset (Merity et al., 2016) consists of articles from Wikipedia. It provides two sizes, WikiText-2 and WikiText-103. They share the same validation and test set, while the training set of WikiText-2 is a truncated version of that of WikiText-103.

**CMU-DoG** CMU Document Grounded Conversations Dataset (Zhou et al., 2018) provides conversation data about the contents of specified documents. Following (Martins et al., 2022), we also use it to test long-range modeling. The dataset contains conversations about movies, the Wikipedia article about the movie is also provided. In this paper, to test the models' ability on modeling long-dependency, we concatenate all conversations to their corresponding Wikipedia articles. The resulting document takes the following form: (WikiArticle,Converstation 1,⋯,Converstation n). Note that the Wikipedia article accounts for only a small portion (1.3%) of a concatenated document on average.

### 4.2 Experimental Methods

Due to limited computational resources, we do not train the language models from scratch. We fine-tune a pretrained language model. We extract the decoder of T5v1.1-LM-adapted [2], and remove the encoder-decoder attention layers from it. It is a version of T5 (Raffel et al., 2020) adapted to LM objective during pretraining. We choose it because it not only uses relative position embedding which is necessary for our purpose, but also is available in a wide range of sizes. Its Small and Base sizes suit our budget. Even though taking only a part of it seems crude, we find it works well on LM tasks in general, even before our fine-tuning.

We fine-tune two sizes of T5v1.1-LM-adapted, namely Small and Base. We simply refer to them as T5 Small and T5 Base. The former has 8 layers and 6 heads, and the latter has 12 layers and 12 heads.

The input size is always $C_{inp} = 512$ tokens. There are two groups of experiments with total cache size of 512 and 640 respectively. In the first group one baseline is pure XL cache of size $C_{xl} = 512$, in other words, same as Transformer-XL (Dai et al., 2019). The other baseline is Compressive Transformer (Rae et al., 2019). We imple-

---

[2]https://github.com/google-research/text-to-text-transfer-transformer/blob/main/released_checkpoints.md

| | XL | 2nd | PG-19 | | WikiText2 | | CMU-DoG | |
|---|---|---|---|---|---|---|---|---|
| | | | Small | Base | Small | Base | Small | Base |
| Vanilla Transformer | - | - | 30.85 | 25.24 | 22.30 | 17.56 | 20.77 | 17.22 |
| XL | 512 | - | 29.89 | 24.38 | 19.85 | 16.31 | 20.15 | 17.19 |
| XL, CPR | 256 | 256 | 29.39 | 24.18 | 19.95 | 16.17 | 20.09 | 17.02 |
| XL, SLC (fix) | 256 | 256 | 29.32 | 24.04 | 19.78 | 16.27 | 19.97 | 16.75 |
| XL, SLC (fix,RSI) | 256 | 256 | 29.46 | 23.99 | 19.77 | 16.19 | 19.83 | 16.63 |
| XL, SLC (slc) | 128 | 384 | 29.08 | 24.14 | 19.13 | 15.65 | 19.12 | **16.11** |
| XL, SLC (slc,RSI) | 128 | 384 | **28.89** | **23.83** | **19.05** | **15.60** | **18.98** | 16.15 |
| XL | 640 | - | 29.62 | 24.43 | 19.77 | 16.27 | 20.23 | 17.41 |
| XL, CPR | 384 | 256 | 29.38 | 24.21 | 19.83 | 16.14 | 20.20 | 17.16 |
| XL, R-SLC | 128 | 512 | **28.50** | **23.63** | **18.57** | **15.20** | **17.95** | **14.82** |

Table 1: Test perplexity on three datasets. Columns "XL" and "2nd" denotes the XL cache size and secondary cache size. "XL", "SLC", "CPR" in rows refer to XL , selective and compressive cache respectively. "(fix)" means using fixation duration as selection criterion. "(slc)" means using selector network. "R-SLC" stands for replacement-based selective cache.

ment their compressive cache in our experimental framework and use the best configuration reported in their paper. For each model, we choose the best cache size configuration between $(C_{xl}, C_{2nd}) = (256, 256)$ or $(128, 384)$, where $C_{2nd}$ is the size of the secondary cache. In other words, given the total size limit, we treat the cache sizes as hyperparameters and search for the best one. In the second group the total cache size is 640 tokens. Because our replacement-based selective cache needs to be greater or equal to the input size, i.e., $>= 512$, and meanwhile, XL cache plays an important role so it is necessary to have a minimum size of it. Thus we use $(C_{xl}, C_{2nd}) = (128, 512)$. For compressive cache, we choose the best configuration between $(C_{xl}, C_{2nd}) = (128, 512)$ or $(256, 384)$, or $(384, 256)$. See more details in Appendix G.

## 4.3 Results

Table 1 shows the results on three datasets (see Appendix H for statistical significance). We can clearly see that using selective cache achieves the best results in all scenarios. Partly replacing XL cache with selective cache results in considerable gains across datasets and model sizes.

We can see that the fixation-based selection surpasses XL cache, especially on PG-19 dataset, which demonstrates the validity of using eye fixation. It achieves similar performance as compressive cache. On the other hand, using the selector network brings about larger improvement, because it allows the model to select what it needs. Note that this is not because of larger selective

cache size, the configuration of cache size shown in the table is the optimal configuration. Moreover, replacement-based selective cache produces even larger improvements. In general, the most substantial performance gains are on the concatenated CMU-DoG dataset. While using XL cache and compressive cache only slightly reduces the perplexity, using selective cache reduces the perplexity by substantial margins. Regarding RSI, it slightly improves the performance in general. It appears to be more effective when the selector is used. It is probably because the key-value pairs selected are more homogeneous in that case.

In addition, the compressive cache shows smaller improvements on PG-19 compared to (Rae et al., 2019), we think it is because of different experimental settings. For example, we fine-tune pretrained models while they train from scratch; we keep the total cache size the same when comparing models; we use smaller models and datasets.

## 4.4 Analysis

**Tokens selected by selector** Figure 3 show the input tokens corresponding to the selected key-value pairs (See Appendix K.1 and K.3 for more examples). It's obvious that the selector has a strong tendency to select named entities. Other than capitalized nouns and some rare tokens, it also selects some normal nouns that are keywords of the text, e.g., "exploration". Note that there is no explicit guidance for this behavior during training. It's interesting that this pattern is automatically learned with a simple LM objective and a $L_0$ penalty. One

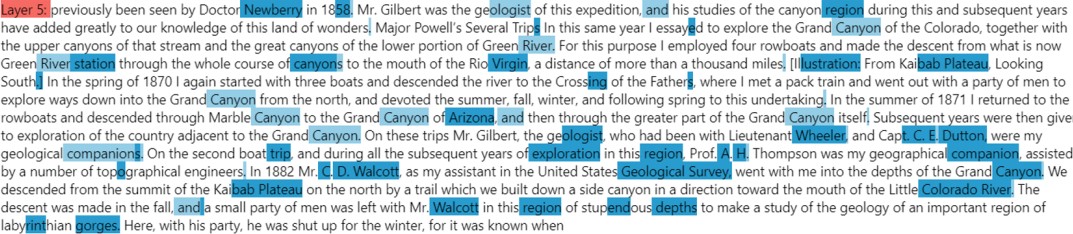

Figure 3: The selected tokens in PG-19 validation set. The *input* tokens of the selected key-value pairs are marked with blue. Deep blue means the key-value pairs pass the secondary selection and are stored in selective cache, while light blue means they are selected by selector but then removed by RSI. The model is T5 Small + XL, SLC (slc,RSI).

possibility is that keeping the named entities in the cache largely facilitates the prediction when the target tokens are themselves. More importantly, it seems that the selector tends to select tokens starting from the second token of a noun phrase. For example, "Doctor <Newberry>" (using "<>" to represent the selection), "Grand <Canyon>" etc. This is a reasonable behavior because when the first token of these noun phrases occurs as the input token, the target is very likely to be the second token. In other words, given that in real data distribution p(2nd token|1st token) is much larger than p(2nd token), so successfully modeling this conditional probability can reduce perplexity significantly. But it is hard without seeing any precedents. Keeping these tokens in cache largely helps to model this kind of conditional probability. From these examples, we can see that the selector tries to make the best use of the cache.

**Effect of RSI**  In the examples shown, RSI seems to work well, at least from the observation of input tokens. The selector repeatedly selects "Canyon". By RSI, only the last one is kept (bottom right corner), and all other "Canyon" tokens are removed. To some extent, it avoids storing similar information in the cache repeatedly, thus saving space for other kinds of representations that carry more diversified information.

**Effect on token-level loss**  We also examine the selective cache from the output side, i.e., the token-level cross-entropy loss. We find that the model makes worse predictions on named entities after masking out part of the selective cache. This, to some extent, explains where the overall perplexity improvements come from. See Appendix K.2.

Combining the findings from both sides, we find the model mainly uses selective cache to store rare tokens or names of characters and places. Impor-

tantly, this retrieval-based working pattern is in line with prior works (Sun et al., 2021; Wu et al., 2022; Hutchins et al., 2022). Therefore, this pattern is not because of the design of selective cache, but rather a general tendency.

**Replacement process**  As for replacement-based selective cache, we calculate the replacement ratio over time. As we expect, we find that the replacement ratio is very high at the beginning of a document, then it quickly decreases and finally stabilizes around a certain level. This is because the density of important key-value pairs in the cache increases over time. Interestingly, on the concatenated CUM-DoG dataset, the replacement ratio stabilizes around zero, which means it preserves the beginning of the document for many steps. Importantly, we observe the same tendency as when using the selector. Most of the newly added tokens in each step belong to named entities. See Appendix L.

**Fixation-based selection**  The fixation duration is predicted by a separate LSTM network. We examine the selected tokens and find that the fixation duration is reasonable in general (see Appendix M). It does not simply stick with a single kind of token such as named entities or rare words. It usually focuses on the core actions in the sentence and the subject and the object of the action, as well as some relevant adjectives and dates. Therefore, it largely reflects human reading behavior (Tokunaga et al., 2017).

Regarding the reason why fixation-based selection is good but not good enough, we think it mainly lies in the discrepancy between the LM task and the task that humans are doing. On the one hand, humans are performing a task of language comprehension when reading. On the other hand, a lookup table for noun phrases could be more advan-

tageous for prediction when they appear multiple times.

## 5 Conclusion

In this work, we propose the selective cache that stores selected key-value pairs from previous context for language models, which significantly increases the length of the context that a language model can attend to. Moreover, we propose three different ways to select important key-value pairs, namely using fixation duration, the selector network and the replacer network. The experimental results from different datasets and model sizes demonstrate the effectiveness of the proposed approaches. Without increasing total cache size, the selective cache outperforms XL cache by considerable margins. Moreover, further analyses reveal important characteristics of the selective cache, such as the neural network-based selection tends to select named entities and rare tokens, as shown by the increased interpretability of our models.

## Limitations

Due to limited computational resources, we evaluate the proposed selective cache by fine-tuning pretrained models on small datasets. In contrast, most of other relevant works train models from scratch on very large language modeling datasets. It is possible that the selective cache performs better or worse in different settings. We cannot directly compare our results with other related work because of the same reason. Moreover, among all the datasets commonly used to evaluate long-range models, we only use PG-19. Other datasets include character-level LM datasets text8 and enwik8 (Mahoney, 2011), source code datasets GitHub (Wu et al., 2022). We have to train from scratch if we evaluate models on those datasets. Finally, while processing long sequences is an ability needed in many applications, we only apply the selective cache on language modeling tasks. The effect of selective cache on other modalities, such as speech and time series is also worth investigating. We hope that future work will continue to evaluate our models on a wider range of datasets and experimental settings.

## Acknowledgements

We acknowledge the computing resources provided at the UCloud platform at SDU eScience Center. We thank the anonymous reviewers for their thoughtful comments on the paper.

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

## A  Eye-tracking Corpora

To train a fixation prediction model, we gather eye-tracking data from four eye-tracking Corpora. Namely, Dundee (Kennedy et al., 2003), GECO (Cop et al., 2017), ZuCo1 (Hollenstein et al., 2018), and ZuCo2 (Hollenstein et al., 2020). We only take the eye-tracking data during natural reading of English text. The collected data comprises multiple domains such as news, novels, movie reviews, and Wikipedia articles. All data is recorded by professional researchers and equipment, with a minimum of 10 subjects reading the same text. There are 1.6 million tokens in total (including repeated text read by different subjects). The total reading time (TRT) of each word is normalized across corpora and averaged over subjects, and then evenly mapped to $\{0, 1, \cdots, 11\}$.

## B  Fixation Prediction Model

We train a fixation prediction model, which uses the same tokenizer and embedding layer as the main transformer language model (the embedding layer is frozen for the sake of generalization on unseen tokens). The model predicts fixation duration on token-level. The original word-level fixation duration is converted to token-level in order to train the model. The conversion is slightly complex: TRT of a word is first assigned to each character of it, then a small number is assigned to the last character of the word (mainly to give small values to punctuation). After tokenizing the word we obtain the span of each subword, and take the maximum value in the span to get the final token-level fixation data. We do it in this way because the tokenizer we are using only provides character-level span information. This is not a perfect solution but it works fine in most cases.

The fixation prediction model consists of an embedding layer (T5 embedding), a two-layer bidirectional LSTM, and a one-hidden-layer MLP on top of it. The best model achieves an MSE of 4.02 on a randomly held-out test set (25% of all data).

## C  Hard Concrete Distribution

The Hard Concrete distribution is based on Binary Concrete distribution (Maddison et al., 2016; Jang et al., 2016), which is defined on the interval $(0, 1)$ and can be regarded as a relaxed version of Bernoulli distribution (see Figure 2). The Binary Concrete distribution is parameterized by $\log \alpha$ and

$\beta$. The location parameter $\log \alpha$ controls the probability mass skewing towards 0 or 1, while $\beta$ controls how sharp the probability density is, or the degree of approximation to a real Bernoulli distribution. The following function can be used to draw samples from this distribution:

$$s = \text{Sigmoid}\left((\log u - \log(1 - u) + \log \alpha)/\beta\right) \tag{9}$$

where $u$ is drawn from a uniform distribution $u \sim \mathcal{U}(0, 1)$, $s$ is the sample drawn from Binary Concrete distribution. To obtain the Hard Concrete distribution, the Binary Concrete distribution is first stretched to $(\gamma, \zeta)$ interval, where $\gamma < 0$ and $\zeta > 1$. It is then rectified to $[0, 1]$.

$$\bar{s} = s(\zeta - \gamma) + \gamma \tag{10}$$

$$z = \min(1, \max(0, \bar{s})) \tag{11}$$

where $z$ is the sample drawn from the Hard Concrete distribution. In the second step, the probability mass between $(\gamma, 0)$ is "folded" to the 0 point, and the probability mass between $(1, \zeta)$ is also "folded" to the point at 1. The stretching and rectifying operations result in non-zero probability at the two endpoints, as well as a continuous curve between them. Figure 2 shows how parameter $\log \alpha$ affects the distribution. For more information about the probability density function and cumulative density function of Hard Concrete distribution, see Appendix B of (Louizos et al., 2017).

When using stochastic masks, the $L_0$ norm is treated in expectation. Then Equation 2 becomes:

$$\mathbb{E}[||z||_0] = \sum_{j=1}^{n} \mathbb{E}[\mathbf{1}_{[\mathbb{R} \neq 0]}(z_j)] \tag{12}$$

$$= \sum_{j=1}^{n} \text{p}(z_j > 0) \tag{13}$$

$$= \sum_{j=1}^{n} (1 - Q_{\bar{s}_j}(0)) \tag{14}$$

where $Q_{\bar{s}_j}(\cdot)$ is the cumulative density function of $\bar{s}$, which is the stretched distribution introduced in Equation 10. The last term is a function of its parameters.

$$\sum_{j=1}^{n} (1 - Q_{\bar{s}_j}(0)) = \sum_{j=1}^{n} \text{Sigmoid}(\log \alpha_j - \beta \log \frac{-\gamma}{\zeta}) \tag{15}$$

In this paper, we follow the recommendation in (Louizos et al., 2017) and use $\beta = 2/3, \gamma = -0.1, \zeta = 1.1$ when training the model with the selector network. Thus the selector only predicts $\log \alpha$.

## D  Training Selector over Previous Snippets

To train the selector, the model maintains a list of previous snippets. Two hyperparameters $l$ and $\text{C}_{snp}$ control the length of the list (the number of snippets stored) and the size of snippets respectively. Note that in this paper, $C_{snp} = 128$. At the end of the current step, a consecutive sub-sequence of hidden states and key-value pairs are randomly selected from all current hidden states $[\vec{h}_1, \cdots \vec{h}_n]$ and all current keys and values $[\vec{k}_1, \cdots \vec{k}_n]$, $[\vec{v}_1, \cdots \vec{v}_n]$. They do not receive gradient as well. If the list is not full, the selected snippet is then appended to the list, otherwise a random old snippet is replaced by it. At the beginning of the next step, one random snippet in the list is selected. The stored hidden states are fed into the selector, and the corresponding keys and values are prepended to the main key and value matrix. The selector produces masks that control the attention to the snippet. Therefore, the selector is trained on a snippet of the previous context, and the time interval between that context and the current input is random. The probability of selecting a snippet from the $k$th previous step is $(1 - \frac{1}{l})^{k-1}\frac{1}{l}$, which is a geometric distribution. Thus $l$ should be set according to the average length of documents.

On the other hand, at the end of each step, the selector selects key-value pairs. But this process does not update the selector. The self-attention layer then learns to attend to whatever is selected. During testing, since the selector does not need training, the list of snippets is always empty, and there are no snippets prepended to the current keys and values.

## E  Architecture of Selector and Replacer Network

The selector network consists of one fully-connected layer, one ReLU activation function (Agarap, 2018), and one fully-connected layer sequentially. The input size, hidden size, and output size are $H, H/4, 1$ respectively, where $H$ is the dimensionality of hidden states of the language model. For each self-attention layer, there is a sepa-

rate selector network, so that the selection criterion is adapted to the need of each layer.

We use the same architecture for the replacer network, with the exception that we add time information. The time interval $\Delta t$ is first converted into a vector and then concatenated with the hidden states. In mathematical form, $\vec{t}_j = \Delta t \cdot [\frac{1}{2^{d_t - 1}} \quad \cdots \quad \frac{1}{2^0}]$, where $d_t$ is the dimensionality for time embedding. In this paper, we use $d_t = 8$. So the input size, hidden size, and output size of replacer is $H + 8, H/4, 1$, where $H$ is the dimensionality of the hidden states.

## F  Detailed Description of RSI

In the case of selector network, suppose that the output $\log \alpha$ for $[\vec{h}_1, \cdots, \vec{h}_n]$ are greater than 0, then their corresponding key-value pairs are selected, i.e., $[\vec{k}_{1,r}, \cdots, \vec{k}_{n,r}], [\vec{v}_{1,r}, \cdots, \vec{v}_{n,r}], \forall r \in \{1, \cdots, m\}$, where $r$ is the index of the head, $m$ is the number of heads. Then a distance matrix $D \in \mathbb{R}^{n \times n}$ is calculated. Each entry $D_{ij}$ is the Euclidean distance between the concatenated values, i.e., $\text{Concate}(\vec{v}_{i,1}, \cdots, \vec{v}_{i,m})$ and $\text{Concate}(\vec{v}_{j,1}, \cdots, \vec{v}_{j,m})$. The reason for choosing values instead of hidden states is that the implementation is simpler. Then the distance matrix is compared with a threshold and converted into a binary matrix $\bar{D}$, whose entries are equal to 1 for distances less than the threshold, and 0 otherwise. Therefore, each row or column represents a collection of similar representations. Then its elements along and above the diagonal are set to be 0, resulting in a strictly lower triangular matrix $\hat{D}$. Finally, the sum of each column is calculated, if $\sum_{i=1}^{n} \hat{D}_{ij} = 0$, then $\vec{k}_{j,r}, \vec{v}_{j,r}$ are remained. If $\sum_{i=1}^{n} \hat{D}_{ij} > 0$, the corresponding key-value pairs are discarded. The additional computation involved in these operations is mainly on the distance calculation. Since this operation is implemented with high-performance code by most popular deep learning frameworks, and the distance is only calculated for selected representations, it does not add much extra computation.

Figure 4 shows the case where $\vec{v}_1, \vec{v}_3$ are similar and $\vec{v}_2, \vec{v}_4, \vec{v}_5$ are similar.

Again, like the coefficient for $L_0$ norm, we find a constant threshold is not appropriate. The overall magnitude of distance varies across different layers. Therefore, we use a threshold that is determined by the average distance. Specifically, threshold $= \eta \frac{1}{n^2} \sum_{ij} D_{ij}$, where $\eta$ is a hyperparameter. Note

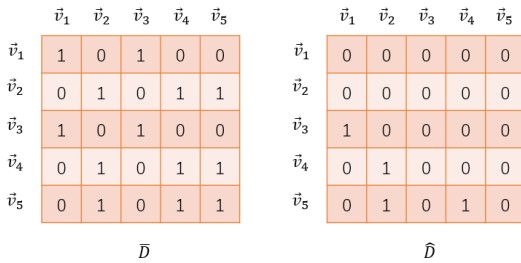

Figure 4: In order to further remove similar representations, a pair-wise Euclidean distance matrix is calculated. It is then converted to a binary matrix $\bar{D}$ whose 1 entries represent that the distance is small enough. It is then converted to a strictly lower triangular matrix $\hat{D}$. If the sum of a column is zero, the corresponding key-value pair is kept.

that in practice, $\eta$ is usually not a small value, so the vectors in the same cluster only bear a limited resemblance.

## G  Implementation Details

**Batching and Processing**  Figure 5 shows how the model is trained with batches of documents. In all of our experiments, the documents are shuffled and then packed in this way. The models process a batch of segments at a time, each segment comes from a different document. The segments are always aligned, i.e., the segments at the same position in the batch dimension are consecutive. Therefore, there are separate caches for each position in the batch dimension. Because the selected number of key-value pairs is different for each cache, we iterate over the batch dimension to update each cache respectively. The cache is emptied at the beginning of each epoch, as well as the beginning and the end of the evaluation. When the model encounters a new document, e.g., from document A to document B in Figure 5, the cache is not emptied. It is unnecessary to do so because the document masks (Section 3) can prevent the model from attending the old document.

During training and validation, the model stops at the shortest row in Figure 5, i.e., at the end of document I in this example, for the sake of efficiency. During testing, the model finishes all the documents, i.e., it stops at the end of document D. The documents are packed in a way such that the rows in Figure 5 have lengths that are as similar to one another as possible. This is important since some documents (especially in PG-19) are

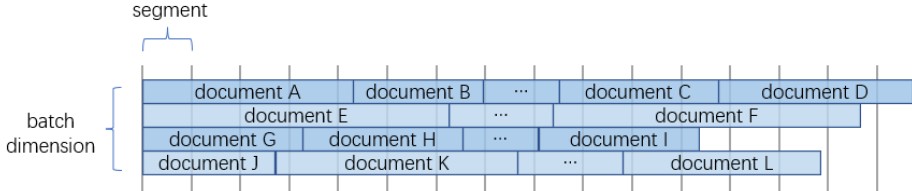

Figure 5: The documents are packed and split into multiple segments. The model processes a batch of segments at a time, and then moves to the next batch of segments (from left to right). In the same position along the batch dimension, i.e., for each row, the segments are continuous. All the models in this paper are trained and tested in this way.

extremely long, and uneven lengths cause a lot of waste.

**Implementation Details**   T5 Small has 8 layers, 6 heads of dimension of 64 in each layer, an embedding size of 512, an FFN hidden layer of size 1024. T5 Base has 12 layers, 12 heads of dimension of 64 in each layer, an embedding size of 768, an FFN hidden layer of size 2048. The two models use GEGLU activation (Shazeer, 2020) in FFN layer. The two models use a sentence-piece (Kudo and Richardson, 2018) tokenizer with a vocabulary size of 32K.

In preliminary experiments, we fine-tune the T5 Small with XL cache and T5 Base with XL cache on truncated PG-19 dataset using learning rate of $\{1 \cdot 10^{-3}, 1 \cdot 10^{-4}, 5 \cdot 10^{-5}, 2 \cdot 10^{-5}\}$, as well as a constant learning rate and a linearly decaying learning rate. We find that a constant learning rate of $1 \cdot 10^{-4}$ works best for both models, when trained for 100K steps. Therefore, we use it for all the experiments presented in this paper. In the following part of this section, unless otherwise specified, the same setting applies to all models. We use AdamW (Loshchilov and Hutter, 2018) optimizer, with weight decay of 0.01. We also use fp16 16-bit mixed precision training. We run all experiments on Tesla T4 GPUs. The models are implemented in Pytorch (Paszke et al., 2019) and based on Huggingface Transformer (Wolf et al., 2020).

For experiments done on the truncated PG-19 dataset, we train the models for 100K steps (59.28 epochs). After 50K steps, the models are evaluated on the validation set every 10K steps, and the best checkpoints are then tested on the test set when the training is finished. We use batch size of 32 for models using T5 Small, and use batch size of 16 and gradient accumulation of 2 steps for models using T5 Base. Note that we train the models on 1% of training data from PG-19 for about 60

epochs instead of training 60% of data for 1 epoch. The reason is that we would like to compare the performance when the models reach convergence. For experiments done on WikiText2 dataset, we train all models that use T5 Small for 50 epochs, with batch size of 32. We train models that use T5 base for 40 epochs (these models converge faster) with batch size of 16 and gradient accumulation of 2 steps. The models are evaluated on validation set every epoch. The best checkpoints are then tested on the test set. For experiments done on the concatenated CMU-DoG dataset, we train all models for 30 epochs, because we find the models start to overfit after 15-20 epochs. we use batch size of 8 and accumulate gradient for 2 steps. We use the same evaluation process as WikiText2.

For compressive cache, we implement the best configuration reported in (Rae et al., 2019), namely, 1D convolution as the compression function trained with attention-reconstruction loss. We follow the compression rate reported in their experiments on PG-19 and use the compression rate of 2. The other experimental settings are the same as other models in this paper, e.g., fine-tuning instead of training from scratch. Recall that we keep track of document IDs to use document masks, the document ID of the first hidden state in the sliding window of convolution operation is assigned as the document ID of the compressed hidden state.

For fixation-based selective cache, we transform the original fixation duration to values ranging from 0 to 11, thus the prediction is also mostly in this range (See preprocessing of eye-tracking data in Appendix A). we then experiment with thresholds of 8, 9, 10 and find 9 works best, which means there are roughly 25% of key-value pairs are selected in each step. Given that $C_{inp} = 512, C_{slc} = 256$, the selective cache only covers two previous steps.

For selector-based selective cache, when training the selector, the size of the context snippet

$C_{snp} = 128$ throughout all experiments, which is removed during evaluation. we haven't tried other values for $C_{snp}$ because we assume it does not affect the performance as long as it is not too small. Besides the cache size, another important hyperparameter is the length of the list that stores previous snippets $l$. Recall that in Section D, the probability of selecting a snippet from the $k$th previous step is a geometric distribution with parameter $l$. In general, It is better to set larger $l$ for modeling longer dependency. We search the best hyperparameter $l$ from choices of $\{3, 5, 7\}$ for PG-19 and Wiki-Text2 datasets, and from $\{6, 9, 12\}$ for CMU-DoG dataset, using a model fine-tuned from T5 small. For PG-19, the best $l = 5$; For WikiText2, the best $l = 3$; For CMU-DoG, the best $l = 9$, and in this dataset the performance is almost the same with different choices. Another hyperparamter is $\lambda'$ in Equation 4. In preliminary experiments, we find $\lambda' = 0.01$ works the best in general among $\{0.03, 0.01, 0.003\}$ and use it for all experiments involving $L_0$ norm. As for the $\eta$ that is used to control the threshold for RSI (See Section 3.4), we simply set $\eta = 0.5$ in all experiments.

As for replacement-based selective cache, we adopt the same hyperparameters when possible. Concretely, I use $C_{snp} = 128, C_{slc} = 512, C_{xl} = 128, C_{inp} = 512$. We also use $l = 5$ for PG-19, $l = 3$ for WikiText2, $l = 9$ for CMU-DoG.

## H  Statistical Significance

We measure the statistical significance of the results. We did 3 runs with different random seeds for 3 configurations, namely T5 Small + XL of size 512, T5 Small + XL, CPR of size (256,256), T5 Small + XL, SLC (slc, RSI) of size (128, 384). We measure the standard deviation of the perplexity as shown in Table 2. Note that we train models on PG-19 dataset for only 10K steps, while we use the same experimental setting on other datasets. We can see the standard deviation is much smaller than the gap between different models.

## I  Training and Inference Cost

We measure the number of parameters, speed and memory usage of different configurations, as shown in Table 3. For all these configurations, we use the same hyperparameters (batch size, model size, etc.) and experiment on the same device (2 Tesla T4) to ensure fair comparisons. We can see the training and inference cost of the proposed

methods are comparable to the compressive cache, and higher than vanilla and XL cache. While these numbers will change on different hardware, we believe the relative proportion will be largely consistent.

## J  Comparison to Simple Baselines

Other than fixation duration and neural networks, we also experiment with two less-sophisticated baselines. (1) The first one ("freq") selects rare tokens. We calculate the token frequency on the training set, and select those tokens whose frequency is lower than the threshold into the selective cache. We determine the threshold so that, on average, around a quarter of tokens in each step are selected (same as when using fixation duration). (2) The second one ("entropy") uses information entropy. When doing auto-regressive LM, the language model estimates the distribution of the next token $P(X_i|x_{<i})$, the entropy of the distribution is $H(X_i) = -\sum_{x_i \in V} P(x_i|x_{<i}) \log P(x_i|x_{<i})$, where $V$ denotes the vocabulary. If the entropy of $X_i$ is large enough, the observed next token will be selected. Intuitively, when the entropy is large, the model is uncertain about the next token, or it's hard to infer from the context, so seeing the next token provides new information. Similarly, we determine the threshold so that the same proportion of tokens are selected in each step.

The experimental results of the new baselines are shown in Table 4. From the table, we can see that the frequency is a better selection criterion than the entropy in general. Compared to Table 1, the frequency and the fixation duration achieve similar performance. On PG-19, the latter performs slightly better, while on CMU-DoG the former performs better. However, the frequency and fixation-based methods demonstrate different selection patterns. E.g., when using frequency as selection criterion, the model selects almost all digits and selects more named entity-related tokens.

Note that the models with selector and replacer networks still show considerable improvements compared to these two baselines. Therefore, even though neural network-based selection tends to select rare tokens, aspects other than frequency also play an important role.

|              | PG-19            | WikiText2        | CMU-DoG          |
|--------------|------------------|------------------|------------------|
| XL           | $34.49 \pm 0.059$ | $19.95 \pm 0.055$ | $20.13 \pm 0.058$ |
| XL,CPR       | $34.09 \pm 0.033$ | $19.92 \pm 0.035$ | $20.13 \pm 0.052$ |
| XL,SLC (slc,RSI) | $33.14 \pm 0.051$ | $19.05 \pm 0.010$ | $18.94 \pm 0.037$ |

Table 2: Mean and standard deviation of 3 runs

|                     | XL  | 2nd | #param | Training | | Inference | |
|---------------------|-----|-----|--------|----------|--------|-----------|--------|
|                     |     |     |        | s/iter | Memory | s/iter | Memory |
| Vanilla Transformer | -   | -   | 52.3 M | 0.71 | 9.0 GB | 0.12 | 2.9 GB |
| XL                  | 512 | -   | 52.3 M | 0.84 | 10.5 GB | 0.14 | 3.1 GB |
| XL, CPR             | 256 | 256 | 56.0 M | 0.95 | 12.4 GB | 0.16 | 3.4 GB |
| XL, SLC (fix)       | 256 | 256 | 52.3 M | 0.93 | 10.5 GB | 0.17 | 3.0 GB |
| XL, SLC (fix,RSI)   | 256 | 256 | 52.3 M | 1.07 | 10.5 GB | 0.20 | 3.0 GB |
| XL, SLC (slc)       | 128 | 384 | 52.3 M | 0.98 | 12.5 GB | 0.16 | 3.0 GB |
| XL, SLC (slc,RSI)   | 128 | 384 | 52.3 M | 1.05 | 12.5 GB | 0.18 | 3.0 GB |
| XL, R-SLC           | 128 | 512 | 52.3 M | 0.98 | 12.2 GB | 0.15 | 3.1 GB |

Table 3: Training and inference cost of difference configurations. "#param" denotes the number of parameters, "s/iter" stands for second per iteration, "Memory" stands for memory usage per GPU.

## K    Analysis of Selector Network

### K.1    Selected Tokens

Other than Figure 3, we show more examples in Figure 6. We can see the same pattern as described previously.

### K.2    Effect on Token-Level Loss

We run two identical models which use selector and selective cache. One model runs normally, while the cached key-value pairs in selective cache of the other model are randomly masked (50% probability). The model cannot attend to those key-value pairs, but those still occupy the room in selective cache. The differences in cross-entropy loss between these two models are then calculated on token-level. In other words, we measure whether the model makes better or worse predictions when predicting each token. Figure 7 shows the resulting loss difference. Note that in previous figures of selected tokens, the color is associated with the input token, while in this figure the color is associated with the prediction target token. We purposely choose the input sequence that is from the same document as the previous example, and a few steps after that. Orange color means the loss increases after masking, green means the loss decreases. Deeper color represents bigger values. The difference values that exceed 1.0 or -1.0 are represented in full color.

### K.3    Difference across Layers

Figure 8 shows examples of the selected tokens in other layers. We can see the selection ratio is different across layers, as well as the "quality" of selection. Besides selecting named entities and rare tokens, no other meaningful patterns are found. The pattern only appears in a few layers. The pattern shown in the figure is consistent in the whole dataset.

### K.4    Performance on CMU-DoG

It is interesting to see how the selector performs in CMU-DoG dataset as shown in Figure 9. Recall that the document in this dataset consists of one Wikipedia article and many conversations based on that article. Therefore, keeping the article in memory is a considerable advantage when doing LM on this dataset.

It is obvious that the selector treats the Wikipedia article and the conversations very differently. The selection ratio drops drastically when coming to the conversation part (in the middle of the third block in Figure 9). This means that the selective cache stores the background article and receives minimal updates afterward. Therefore, the background article can be preserved for many steps. Note that this beneficial pattern is learned by the model itself, and there is no token type embedding or something similar to explicitly distinguish two types of inputs. Furthermore, this example reveals a key difference between the selective cache and

| | XL | 2nd | PG-19 | WikiText2 | | CMU-DoG | |
|---|---|---|---|---|---|---|---|
| | | | Small | Small | Base | Small | Base |
| XL, SLC (freq) | 256 | 256 | 29.53 | 19.86 | 16.25 | 19.85 | 16.66 |
| XL, SLC (entropy) | 256 | 256 | 29.91 | 19.88 | 16.24 | 20.24 | 17.02 |

Table 4: Test perplexity when using frequency ("freq") and information entropy ("entropy") as selection criterion.

Figure 6: The selected tokens in WikiText2 validation set. See the caption of Figure 3 for an explanation of colors. The model is T5 Small + XL, SLC (slc,RSI).

Figure 7: Difference in token-level loss when randomly masking 50% key-value pairs in selective cache in layer 4,5 and 6. Orange color means the loss increases after masking, green means the loss decreases. The deeper color represents bigger values. The model is T5 Small + XL, SLC (slc, RSI) trained on PG-19.

Figure 8: The selected tokens in PG-19 validation set. See the caption of Figure 3 for an explanation of colors.

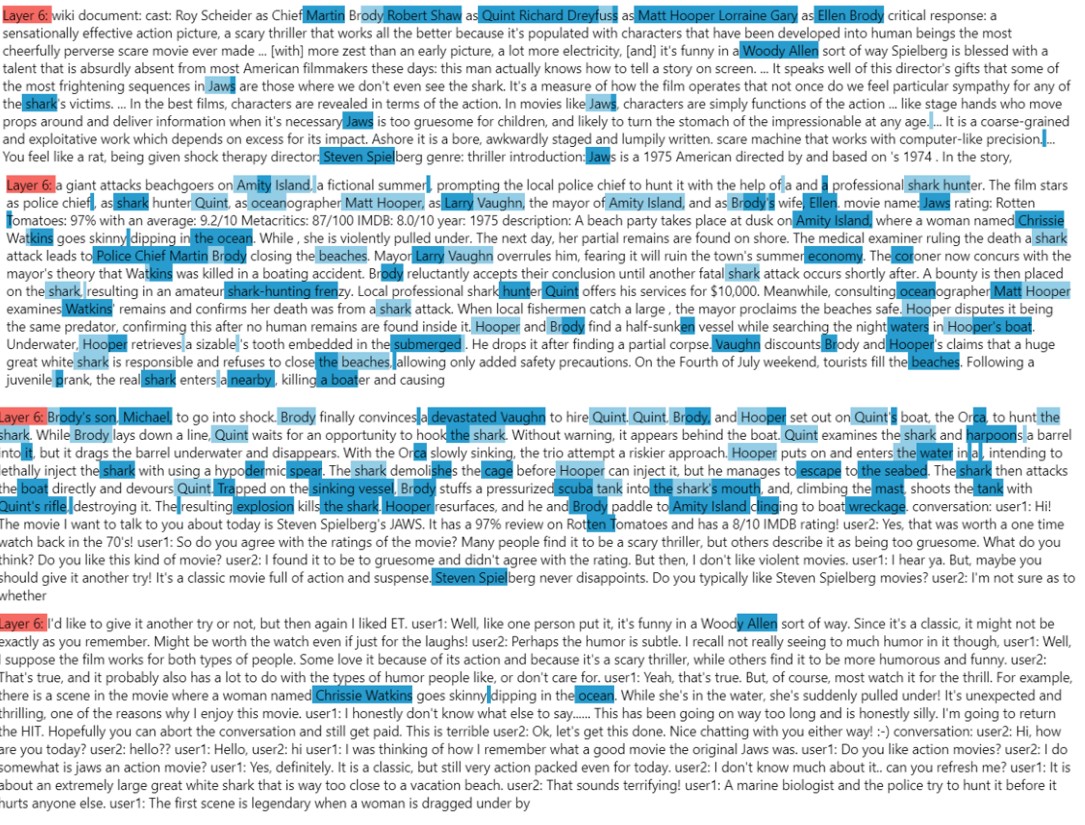

Figure 9: The selected tokens in the concatenated CMU-DoG valid set. The input sequences shown are consecutive, starting from the beginning of a document. See Figure 3 for the explanation of colors. The model is T5 Small + XL, SLC (slc,RSI).

compressive cache. The latter defines a fixed compressive rate beforehand, while selective cache is adaptive to the varying density of information in the input and has varying compressive rates.

In addition, different from previous examples on PG-19 and WikiText2, in this example the selector not only selects named entities or rare words, but also selects some common words such as "shark", "tank", even including a verb "escape". This shows that the selection is not necessarily restricted to named entities or rare words. The selector can further learn to select other kinds of tokens such as keywords.

## L  Analysis of Replacer Network

### L.1  Analysis of Replacement Process

Figure 10 shows how the replacement ratio changes over time. The peaks in the figure correspond to the start of a new document. It seems that the selective cache can preserve items for infinite steps on CMU-DoG, since there are some nearly zero replacement ratios. We found this is the case for most layers.

Figure 11 shows an example of replacement. Both new key-value pairs and the cached ones are shown. They are aligned to show that "shops" and "," are compared (more precisely, their corresponding inner representations), "districts" and "@" are compared, etc. Note that the "cache" rows contain non-continuous context, while "new" rows contain a continuous input sequence.

## M  Tokens Selected According to Fixation Duration

Figure 12 shows the selected token according to fixation duration. we can see the differences compared to those selected by the selector network. Humans are doing comprehension when reading, such as building the relationship map between characters and understanding the plot, and the long-fixated tokens are key information for that. As we can see, an important difference is that verbs compose a substantial proportion in those tokens selected by fixation duration, while they are excluded by the selector network. We think those verbs are also helpful for predicting future tokens, but that requires higher-level abilities. It is possible that when combined with a more advanced or much larger language model which has the capacity to do complex reasoning and comprehension over previous distant context, the fixation-based selection can bring about much larger performance gains.

Meanwhile, another important advantage of using fixation duration is that it does not require training or fine-tuning together with the language model. So it can be used to select key-value pairs for those very large pretrained language models that are too expensive to calculate gradients.

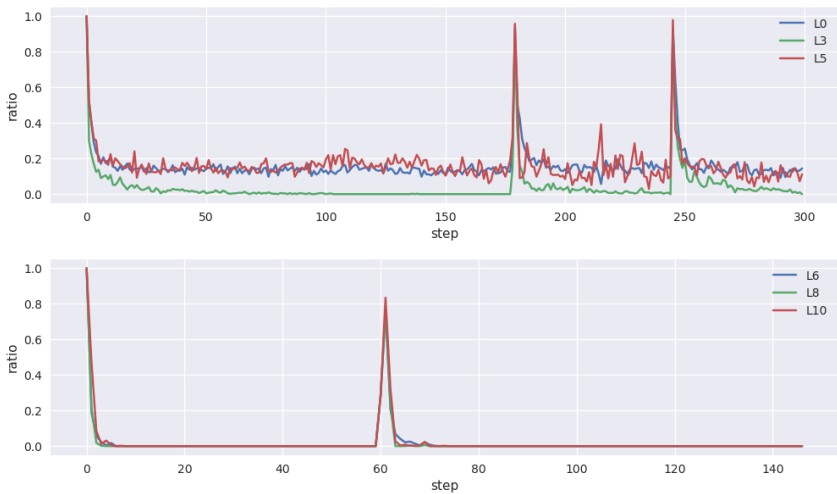

Figure 10: Replacement ratio over time. "L*x*" denotes the layer number. **Top**: T5 Small + XL, R-SLC on PG-19 dataset. **Bottom**: T5 Base + XL, R-SLC on the concatenated CMU-DoG dataset.

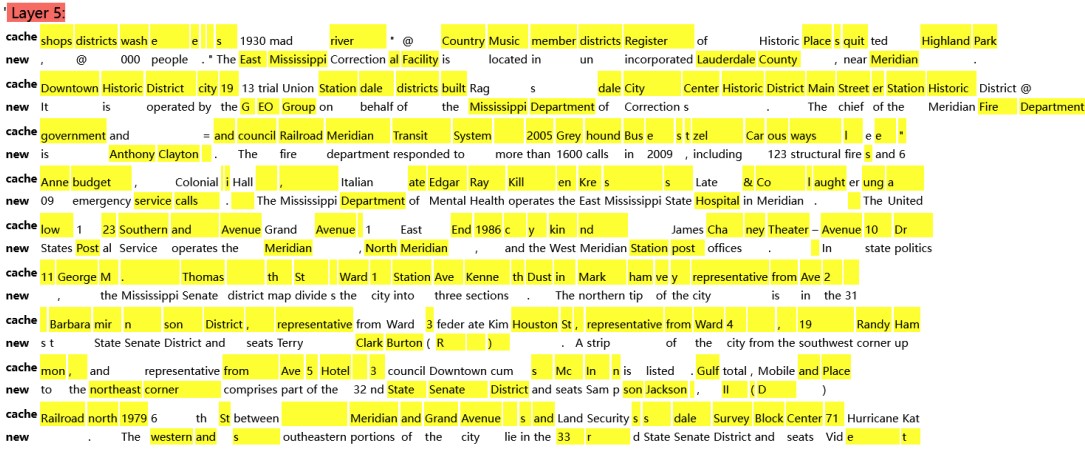

Figure 11: A example of replacement. Rows starting with "cache" depict the content of the selective cache, while rows starting with "new" show the new candidates from the input sequence. Tokens that are aligned to represent the one-to-one relationship. Highlighted tokens mean that their corresponding key-value pairs are kept after the replacement process. The configuration is T5 Small + XL, R-SLC on the WikiText2 dataset.

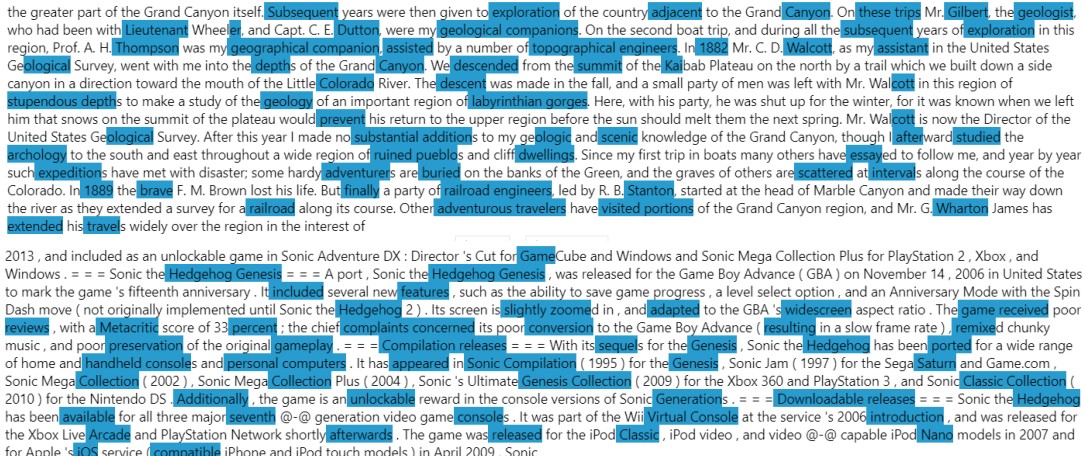

Figure 12: **Top:** The selected tokens on PG-19. **Bottom:** The selected tokens on WikiText2. Those long-fixated tokens are selected, based on the fixation duration predicted by a separate model. The *input* tokens of the selected key-value pairs are marked with blue. The RSI is not used here.