# OpenReview forum: "Long-Range Language Modeling with Selective Cache"
_EMNLP/2023/Conference — EMNLP 2023 Findings_

### Official Review · Reviewer_WTsa · 2023-08-03

**Soundness:** 3

**Excitement:**

4: Strong: This paper deepens the understanding of some phenomenon or lowers the barriers to an existing research direction.

**Paper Topic And Main Contributions:**

This paper proposes three strategies to build a selective cache of key-value pairs (or "states") in a transformer XL style language model which enables the model to  make more effective use of longer histories using a fixed size cache. The strategies include 1) selecting words based on eye-fixation duration, 2) using a selector network which learns to prune key-value pairs and 3) a refinement of the selector network called replacement network which learns to remove key-value pairs from the cache. Results on PG-19, Wikitext and CMU-DoG datasets show that all 3 strategies yield improvements over Transformer XL and Compressive Cache (Rae et al. 2019). The replacement network strategy performs the best. Additionally the paper reports small improvements by clustering similar tokens and selecting only one key-value pair from each cluster.

Strengths:
* Proposes 3 novel approaches to improve the effectiveness of a transformer LM by caching important key-value pairs without increasing the context over which attention is computed.
* Results on three tasks show that the proposed approach yield perplexity improvements over transformer XL  (Dai et al. 2019) and compressive cache (Rae et al. 2019)
* Demonstrates that the approaches tend to store named entities and rare tokens in the cache, an intuitive result.

Weaknesses:
* Some experimental comparisons are weak/incomplete. See comments below

Comments:
* In Table 1, the results from selector network (slc) do not use the same XL and secondary cache size as the SLC (fix). While the total cache size (XL+2nd) for slc is the same as the baseline (512), it would have been preferable if the slc used the exact same cache sizes for XL/2nd. This is also the case for the replacement network (R-SLC).
* In Table 1, the result from the replacement network is not reported for the cache size of 512.


**Questions For The Authors:**

* In Section 3.2, total reading time is at the word level but the model operates at a subword (wordpiece/sentencepiece) level - how do you deal with this aspect?


**Reasons To Accept:**

* Given the immense popularity of large language models in the NLP community, the techniques proposed in this work can help improve performance of language models while lowering computation, which is meaningful and will likely benefit several researchers without access to compute resources.


**Reasons To Reject:**

* Some of the experimental comparisons in the paper are not fully convincing. See comments above.

**Reproducibility:**

4: Could mostly reproduce the results, but there may be some variation because of sample variance or minor variations in their interpretation of the protocol or method.

**Reviewer Confidence:**

5: Positive that my evaluation is correct. I read the paper very carefully and I am very familiar with related work.

---

> ### Author Rebuttal · Authors · 2023-08-28
>
> Thank you so much for your review!
>
> * About your question
>
>     We convert the word-level TRT to subword level in a slightly complex way: For example, "biggest, (6)" represents a word "biggest," and its Total Reading Time 6 (after converting to integers).
>
>     "biggest, (6)"
>
>     TRT is first assigned to each character, resulting
>
>     "b (6)", "i (6)", ... ", (6)".
>
>     Then assign a small TRT to the last character,
>
>     "b (6)", "i (6)", ... ", (1)".
>
>     After tokenizing the word we obtain the span of each subword,
>
>     "big", "g", "est", ","
>
>     And take the maximum value in the span.
>
>     "big max(6,6,6)", "g max(6)", "est max(6,6,6)", ", max(1)"
>
>     The final result is
>
>     "big (6)", "g (6)", "est (6)", ", (1)"
>
>     We do it in this way because the tokenizer we are using only provides character-level span information. This is not a perfect solution but it works fine in most cases.
>
> * About your comments
>     * In Table 1, we use different XL and secondary cache sizes for different configurations. As mentioned in the paper, we experiment with (256, 256) and (128, 384) for each configuration and choose the best one to report. In general, we find the perplexity of SLC (fix) 128 384 > SLC (fix) 256 256 > SLC (slc) 256 256 > SLC (slc) 128 384. If we choose the same cache size for all configurations, then the difference between SLC (fix) and SLC (slc) will depend on our choice. (256, 256) produces a smaller difference while (128, 384) produces a larger difference. So we treat the cache sizes as hyperparameters and search for the best one for each configuration.
>     We do the same when comparing R-SLC to CPR, we search the best hyperparameter from (128, 512), (256, 384), (384, 256) for CPR, while using (128, 512) for R-SLC.
>
>     * When experimenting with the replacer network, as mentioned in the paper, the R-SLC cache size needs to be greater or equal to the input size, i.e., >= 512. Meanwhile, XL cache plays an important role so it is necessary to have a minimum size of it. Thus we use (128, 512) for R-SLC and we have to run the baseline models with the total cache size of 640. Using (0, 512) for R-SLC would result in bad performance.

---

### Official Review · Reviewer_ziVP · 2023-08-03

**Soundness:** 3

**Excitement:**

3: Ambivalent: It has merits (e.g., it reports state-of-the-art results, the idea is nice), but there are key weaknesses (e.g., it describes incremental work), and it can significantly benefit from another round of revision. However, I won't object to accepting it if my co-reviewers champion it.

**Paper Topic And Main Contributions:**

This paper extends upon the idea to cache relevant information from previous context to enable the learning long-range dependencies in language models.

Specifically, on top of the usual XL cache, the authors propose to include an additional secondary selective cache, which by some selection criterion, is populated by long-range relevant key-value pairs. They also proposed a differentiable proxy to overcome the non-differentiable nature of the selective cache. Three selection criteria are considered in this paper, namely, human eye-fixation duration on different words as predicted by a pre-trained LSTM network, a dedicated selector network, and a replacer netowrk.

The topic is of interest for EMNLP, because modeling long-range depencies remain a challenging direction to further improve the LMs.
The authors showed experimental results, that with their method, PPL can be improved on top of the vanilla transformer, as well as the XL model. Additionally, the authors provided detailed analyses to support that the selection process actually kept important words into the cache.

**Questions For The Authors:**

There are two main questions from my end:

1. Why don't you compare to some less sophisticated and easer-to-enderstand baselines? For example, simply using the information entropy as the selection criteron.

2. You mentioned that a pre-trained T5 model is used and further adapted. My question is, how comparable is the data condition then for different rows in Table1? I feel adding a column for "data condition" etc. would help with the clarity.

**Reasons To Accept:**

The proposed method is of sufficient interest. Such a cache which essentially prunes irrelevant information while keeping important information from the history is an interesting direction to enable long-range modeling capabilities of the LMs.

Decent PPL improvements are obtained with the proposed method.

**Reasons To Reject:**

The proposed method is rather complicated, which requires pre-training the LSTM eye-fixation duration prediction network, the selector or the replacer network.

Although qualitatively explained, the quatitative improvements from the proposed method remains somewhat unclear to me - yes we saw PPL improvemtns, but where do they really come from? Is it really like expected, that the improved PPL can be explained,

**Reproducibility:**

4: Could mostly reproduce the results, but there may be some variation because of sample variance or minor variations in their interpretation of the protocol or method.

**Reviewer Confidence:**

3: Pretty sure, but there's a chance I missed something. Although I have a good feel for this area in general, I did not carefully check the paper's details, e.g., the math, experimental design, or novelty.

**Typos Grammar Style And Presentation Improvements:**

Eq. 2 is kind of "floating" on its own, I felt the smoothness of the reading experience was broken abruptly when I first reached L321. Maybe some connection sentences to explain what it is could help with the problem.

---

> ### Author Rebuttal · Authors · 2023-08-28
>
> Thank you so much for your review! Here is our reply to your questions:
>
> * Comparison to less-sophisticated baselines
>
>     As you suggested, we experiment with two simpler methods.
>
>     (1) The first one ("freq") selects rare tokens. We calculate the token frequency on the training set, and select those tokens whose frequency is lower than the threshold into the selective cache. We determine the threshold so that, on average, around a quarter of tokens in each step are selected (same as when using fixation duration).
>
>     (2) The second one ("entropy") uses information entropy. When doing auto-regressive LM, the language model estimates the distribution of the next token $P(X_i|x_{<i})$, the entropy of the distribution is
>     $ H(X_i)  = - \sum_{x_i \in V} P(x_i|x_{<i}) \log P(x_i|x_{<i}) $
>
>     where $V$ denotes the vocabulary. If the entropy of $X_i$ is large enough, the observed next token will be selected. Intuitively, when the entropy is large, the model is uncertain about the next token, or it's hard to infer from the context, so seeing the next token provides new information. Similarly, we determine the threshold so that the same proportion of tokens are selected in each step.
>
>     A random sample of the selected token based on entropy (parentheses indicate selection): Kyra was (created) as a (separate) character from (the) (Se)er , who was (featured) as (the) (primary) antagonist in the (second) half of season four . (Carpenter) joked that she was surprised to be (considered) for such a minor part on the show as a (guest) star .
>
>    The experimental results of the new baselines are as follows:
>
>     | configurations | PG-19, Small | WikiText2, Small | WikiText2, Base | CMU-DoG, Small | CMU-DoG, Base |
>     | ------------- | ------------ | ---------------- | --------------- | -------------- | ------------- |
>     | XL,SLC (freq) 256 256 |29.53 |    19.86         |      16.25      |    19.85      |     16.66     |
>     | XL,SLC (entropy) 256 256 |29.91 |   19.88       |     16.24     |    20.24      |     17.02     |
>
>     From the table, we can see that the frequency is a better selection criterion than the entropy in general. Compared to Table 1 in the paper, the frequency and the fixation duration achieve similar performance. On PG-19, the latter performs slightly better, while on CMU-DoG the former performs better. However, the frequency and fixation-based methods demonstrate different selection patterns. E.g., when using frequency as selection criterion, the model selects almost all digits and selects more named entity-related tokens.
>
>     Note that the models with selector and replacer networks still show considerable improvements compared to the new baselines. Therefore, even though neural network-based selection tends to select rare tokens, aspects other than frequency also play an important role.
>
> * Data condition for different rows in Table 1
>
>     The data condition is the same for all rows in Table 1. The original T5v1.1 model is further adapted to LM objective, producing a new version T5v1.1-LM-adapted, released by google research. All our experiments are fine-tuning from this version, so the data condition is the same.

---

### Official Review · Reviewer_voB6 · 2023-08-04

**Soundness:** 3

**Excitement:**

3: Ambivalent: It has merits (e.g., it reports state-of-the-art results, the idea is nice), but there are key weaknesses (e.g., it describes incremental work), and it can significantly benefit from another round of revision. However, I won't object to accepting it if my co-reviewers champion it.

**Missing References:**

Fine

**Paper Topic And Main Contributions:**

The topic of this paper is about long-range language modeling. This paper introduces the selective cache, which stores the selected key-value pairs from the previous context. It proposes two selection methods, the first one is based on eye-fixation duration, and the second one is based on training. And it proposes a method to remove similar instances for further improvement. Experiments are conducted on three datasets, including the subset of commonly used datasets for long-range language modeling.

**Questions For The Authors:**

A. Why choose the T5 decoder, which removes encoder attention instead of a natural decoder-only pre-trained model?
B. Can you give more analysis or demonstration about the training cost as well as the inference cost? As you mentioned, the limitation of computational resources, is this method training heavily?

**Reasons To Accept:**

This paper proposes a sophisticated method for long-range language modeling. The improvement is obviously compared with Vanilla Transformer, Transformer-XL and CPR. The selective cache is well-designed with two insightful selection criterions. The case analysis is interesting.

**Reasons To Reject:**

Experiments are limited. The paper didn't compare with other recently related work because of the different settings, as it only conducts experiments on a small dataset or subset.
The method is complex, and the effect of each design on performance gains is hard to attribute. The analysis experiments of the necessary design are needed.

**Reproducibility:**

3: Could reproduce the results with some difficulty. The settings of parameters are underspecified or subjectively determined; the training/evaluation data are not widely available.

**Reviewer Confidence:**

2: Willing to defend my evaluation, but it is fairly likely that I missed some details, didn't understand some central points, or can't be sure about the novelty of the work.

**Typos Grammar Style And Presentation Improvements:**

The paper needs more introduction about the compared baselines, and the contributions should be clarified more obviously.

---

> ### Author Rebuttal · Authors · 2023-08-28
>
> Thank you so much for your review! Here is our reply to your questions:
>
> * Why use T5 decoder?
>
>     Because we need pretrained language models with relative position embeddings, we intended to use Pretrained Transformer-XL (Dai et al, 2019), but the available sizes of Transformer-XL are too large. On the other hand, T5 is available in a wide range of sizes, its Small and Base size suit our budget. Even though taking only a part of it seems crude, we find it works well on LM tasks in general, even before fine-tuning. We will add a short justification for the choice of the model in the next version of the paper.
>
> *   Analysis about the training and inference cost
>
>     |   configurations  | #param | s/iter (train) | Memory (train) | s/iter (infer) | Memory (infer) |
>     | ------ | ------ | -------------- | -------------- | -------------- | -------------- |
>     | Vanilla | 52.3 M| 0.71           | 9.0 GB         | 0.12           | 2.9 GB         |
>     |XL 512   | 52.3 M| 0.84           | 10.5 GB        | 0.14           | 3.1  GB        |
>     |XL,CPR 256 256 | 56.0 M | 0.95      | 12.4 GB        | 0.16           | 3.4 GB         |
>     |XL,SLC (fix) 256 256| 52.3 M | 0.93   | 10.5 GB        | 0.17           | 3.0 GB         |
>     |XL,SLC (fix,RSI) 256 256| 52.3 M | 1.07 | 10.5 GB      | 0.20           | 3.0 GB         |
>     |XL,SLC (slc) 128 384| 52.3 M | 0.98  | 12.5 GB        | 0.16           | 3.0  GB        |
>     |XL,SLC (slc,RSI) 128 384| 52.3 M | 1.05 | 12.5 GB      | 0.18           | 3.0 GB         |
>     |XL,R-SLC 128 512 | 52.3 M | 0.98    | 12.2 GB        | 0.15           | 3.1 GB         |
>
>
>     We measure the number of parameters, speed and memory usage of different configurations, as shown in the table above. For all these configurations, we use the same hyperparameters (batch size, model size, etc.) and experiment on the same device (2 Tesla T4) to ensure fair comparisons. “s/iter” is the time for each iteration or step, “Memory” is the memory usage per GPU. We can see the training and inference cost of the proposed methods are comparable to the compressive cache, and higher than vanilla and XL cache. While these numbers will change on different hardware, we believe the relative proportion will be largely consistent. We will add this information in the appendix of the paper.
>
> Regarding your concern about limited experiments, we can provide additional results. After submission of the paper we continue to run experiments using T5 Base on PG-19 dataset, here is the result. Note that the conclusions of the paper are unaffected. We will add this to Table 1 in the next version of the paper.
>
> |   configurations  | PG-19, Base |
> | ------ | ------ |
> | Vanilla | 25.24|
> |XL 512   | 24.38|
> |XL,CPR 256 256 | 24.18|
> |XL,SLC (fix) 256 256| 24.04 |
> |XL,SLC (fix,RSI) 256 256| 23.99 |
> |XL,SLC (slc) 128 384| 24.14 |
> |XL,SLC (slc,RSI) 128 384| 23.83 |
> |XL 640   | 24.43|
> |XL,CPR 384 256 | 24.21|
> |XL,R-SLC 128 512 | 23.63 |

---

### Meta-Review · Area_Chair_BNJy · 2023-09-29

**Recommendation:** 3

**Metareview:**

Paper proposes an efficiency improvement method for long-range transformers. While the reviewers found the proposed method interesting, they also highlighted the complexity of it and questioned whether the conducted experiments were enough to explain the complexity of the proposal.

---

### Decision · Program_Chairs · 2023-10-07

**Decision:**

Accept-Findings

**Comment:**

Paper proposes an efficiency improvement method for long-range transformers. While the reviewers found the proposed method interesting, they also highlighted the complexity of it and questioned whether the conducted experiments were enough to explain the complexity of the proposal.